# The Collaboration of Private Hospitals with the Public Health Service: The Case of La Rioja, Spain (1986–2019)

**DOI:** 10.3390/healthcare12100990

**Published:** 2024-05-11

**Authors:** María Teresa Jiménez-Buñuales, Pilar León-Sanz, Paulino González-Diego, Leonor González-Menorca

**Affiliations:** 1Preventive Medicine Unit, Hospital of Calahorra, Riojan Health Service, 26500 Calahorra, Spain; 2Department of Biomedical Humanities, Faculty of Medicine, University of Navarra, 31008 Pamplona, Spain; mpleon@unav.es; 3Independent Researcher, 31009 Pamplona, Spain; paulinogonzalezdediego@gmail.com; 4Faculty of Economics and Business Sciences, University of La Rioja, 26006 Logrono, Spain; leonor.gonzalez@unirioja.es

**Keywords:** National Health Service, public hospitals, private hospitals, La Rioja, Spain, twentieth and twenty-first centuries, providers of systems of care

## Abstract

In Spain, the public National Health Service provides care to Spaniards and other residents and is tailored for a decentralized state of autonomies. Each Autonomous Community has legislative capacity in its organization and management. We study the case of the collaboration between private hospitals and the public health service in La Rioja, an Autonomous Community of Spain located in the North of the Iberian Peninsula, due to the importance that this relationship has in health systems, in general. We applied the case study method as a methodological tool in a long-term local study. The interpretation was carried out within a national context, which allows us to understand its meaning and the historical keys to hospital development in this region. Primary sources have been reviewed (mainly reports, catalogs, and censuses of hospitals from the Ministry of Health and the Government of La Rioja) and other secondary sources, located in archives, libraries, Institute of Rioja Studies, and Department of Health. The hospital system in La Rioja was characterized by a predominance of public beds compared with private ones, although there has been a growing trend in the number of private beds from 2013 onwards due to the incorporation of health and social care convalescent hospitals (two). La Rioja has been promoting public–private collaboration (seen as a strategic alliance) and focusing on agreements in the socio-health space, particularly using the management service agreement and the concession of work formulas. The development of the public health service in La Rioja, from 1986 to 2019, has been determined by a progressive lower dependence on specialized hospitals from other health services of neighboring Autonomous Communities and by a mixed public–private hospital system.

## 1. Introduction

Hospital care is an essential part of the healthcare system. The study of the historical development of hospital systems and the dependence on private and public hospital infrastructure for care provision has become a topic of renewed interest in international academic circles due to its relevance to healthcare and the economic cost it entails [1] The National Health Service, which serves almost all Spaniards and residents in Spain, is the result of a complex historical process and has transitioned from a social security-based model financed by contributions to a tax-funded system inspired by national health services [2]. 

At this moment, the National Health Service is characterized by universal coverage, public financing, and the predominantly public provision of services, which aims to guarantee access to the provision of health services. 

The Spanish health service is a system fully decentralized in regions termed Comunidades Autónomas (Autonomous Communities). Each Autonomous Community has legislative capacity for its organization and management. In Spain, the delegation of functions in the field of health to the regional governments of the 17 Autonomous Communities has been implemented progressively, between 1981 and 2001 [3].

La Rioja is an Autonomous Community of Spain located in the North of the Iberian Peninsula with a population of 46,934,632 (1 January 2019). Spain covers 505,955 km^2^; it is the third largest area in western Europe. La Rioja has an area of 5045 km^2^, so it is a small region compared to the whole of Spain, with a population of 315,931 inhabitants. In the analysis of demographic evolution in La Rioja, the two most relevant features of the Riojan population are aging and immigration [4]. The evolution of these sociodemographic characteristics has influenced the changes in the patient profile in La Rioja, which includes older age, greater comorbidity and complexity, chronicity, and dependence [5]. 

In the regulatory framework, it is worth highlighting the application of three regulations: Firstly, the regulation in Law 14/1986 of 25 April, General Health Law, which established the National Health Service, which encompasses the health services provided by the State Administration and the Health Services of the Autonomous Communities [6] indicated that Spanish public health services could establish agreements with private hospitals. In general, they were, and are, long-term agreements, which reflect mutual responsibilities in promoting common interests [7]. Secondly, Royal Decree 1473/2001 of 27 December transferred health responsibilities and health services from the National Institute of Health to La Rioja [8]. Thirdly, Law 2/2002 of 17 April defined the Public Health Service of La Rioja as the set of organizational means, resources, and actions to guarantee the right of people to health protection [9]. The Riojan Health Service, established in 1991, is the public entity dependent on the Ministry for Health of the Government of La Rioja, responsible for overseeing healthcare services and benefits provided by public health facilities in La Rioja. 

To define the term Public-private partnerships (PPP) applied to the health sector, this article takes into account Jütting’s idea about the health system, the provision of services, and the measurement systems of the possible contributions of the private sector to the public sector used by this author, referring to efficiency, to the access to services, and the production of additional resources and income for the public sector [10]. Koppenjan points out that they comprise a type of structured cooperation between public and private partners in the planning/construction and/or operation of infrastructure, in which they share or redistribute risks, costs, benefits, resources, and responsibilities [11]. The National Council of Public-Private Partnerships of the United States of America highlights the association of the public sector and private for-profit companies, through which the infrastructure of the public service is increased [12].

Until now, Spanish historiography on the history of hospitals lacks comprehensive and long-term interpretations that allow us to understand the collaboration between public health services and private hospitals.

This research aims to study the collaboration of private hospitals with the public health service from a territorial perspective, through the case of La Rioja, describing the origin and evolution of private hospitals, and considering their ownership and the impact that the collaboration had on both private hospitals and the public health service. We will review key elements of the process and the influence of this co-collaboration on the modernization and evolution of the care provided [13,14]. The case of La Rioja, for the period 1986–2019, serves as a case study because it reflects what happened in other regions.

In this article, after reviewing the sources and the methodology used, we will focus on the presentation of the data of the private hospitals hired by the National Public Health Service and by the Riojan public health service. We also present the characteristics of Riojan private hospitals in order to understand the type of collaboration established and the influence that this collaboration had. We refer to the types of hospital agreements established. We focus the Discussion section on the evolution (1986–2019) of private hospitals in La Rioja and in the characteristics and the impact of public-private collaboration for the public health service of La Rioja in this period.

## 2. Sources and Method

A local study and an interpretation were carried out on a long-term basis and within a national context to analyze the development of private hospital supply and its relationship with public hospital resources, which is crucial for understanding the hospital development of La Rioja. We have applied the case study method as a methodological tool in this research. We revised the framework of hospitals in La Rioja and each hospital’s regulation and specific sources. Additionally, we analyzed data, comparing hospitals in La Rioja with those in other Autonomous Communities within the National Health Service. 

The case study method applied aims to provide a view of an emerging event that is examined to understand it comprehensively, and determine the elements that condition it and the possible associated consequences. With this case study method, the research pretends to understand the process by which certain phenomena occur [15]. Remarkable characteristics of this methodology for the article are as follows: (i) it investigates a contemporary phenomenon in its environment, (ii) reveals that the boundaries between the phenomenon and its context are not clearly evident, and (iii) uses multiple sources of data [16]. 

Three main moments are considered: heuristic (documentation search), hermeneutic (the interpretation of data in its context), and critical (the evaluation of documentation). Primary and secondary documentary sources from the National Health Management Institute archives and the Government of La Rioja, including the General Archives of La Rioja [17,18] and the Institute of Rioja Studies, are reviewed. Municipal archives from La Rioja, Logroño [19], Lardero [20], Haro [21], and Calahorra [22,23] are also studied. Information from hospital catalogs and censuses from the Department of Health and the Ministry of Health [24,25], data from the National Institute of Statistics and the Institute of Statistics of La Rioja, and other documents, such as institutional reports, are also utilized. 

This study begins with 1986, the date of the approval of the General Health Law, and ends in 2019 to delimit a period of homogeneous healthcare activity without the particularities derived from hospital care during the exceptional situation experienced during the coronavirus SARS-CoV-2 (COVID-19) pandemic.

## 3. Results

The public network of the National Health Service comprises publicly funded and private hospitals that have a substitute agreement or belong to a public utilization network; that is, they are funded with public funds. Throughout the 1986–2019 period, the proportion of public hospitals was 56.22% (on average) in La Rioja and 44.62% in Spain. At the beginning of this study (1986), La Rioja had seven hospitals, of which 43% were private; this number remained unchanged at the end of the study period. In 2000, the proportion of public hospitals increased to 80%, and that of private hospitals decreased to 20%. This pattern changed in La Rioja in 2013 (public hospitals: 39.67%; private hospitals: 55.13%) (Figure 1). Likewise, the proportion of public beds throughout the 1986–2019 period was 87.75% (La Rioja) and 68.25% (Spain as a whole) on average, and the proportion of private beds was 12.25% (La Rioja) and 31.75% (Spain as a whole) on average. From 2013, there was a growing trend in the proportion of private beds (22%), and the proportion of public beds decreased to 78% (a more pronounced pattern than that at the national level) (Figure 2). At the end of this study (2019), 57% of hospitals were private.

Hospitals in La Rioja during the study period (1986–2019) are segmented by type: general hospitals, psychiatric hospitals, and convalescence hospitals (Table 1). Private hospitals included Sanatorio Velázquez, Policlínico Quirúrgico Nuestra Señora del Carmen, Policlínico Riojano Nuestra Señora de Valvanera, Clínica Los Manzanos, Convalescence Hospitals Los Jazmines, and Nuestra Señora Virgen del Carmen. In addition, the Calahorra Hospital Foundation, a publicly managed hospital that was established in 2000, is included. This hospital adhered to the non-profit healthcare organization model outlined in Royal Decree 29/2000 of 14 January, which introduced new management methods for the National Health Institute (INSALUD), the former public entity responsible for the healthcare provision and management until the current National Health Service was configured. After the study period and according to Law 1/2022 of 23 February, the Calahorra Hospital Foundation was transformed into a public health Foundation and became part of the Riojan Health Service.

### 3.1. Sanatorio Velázquez

Sanatorio Velázquez or Velázquez Clinic was owned by the Sociedad Anónima Velázquez and was established in 1965. It was a privately owned center located in Logroño that contained approximately fifty beds and had an agreement with the public Health Service of La Rioja. It was a general hospital with departments for medicine, surgery, obstetrics and gynecology, and pediatrics (classified in Group IV, surgical, type IA, the first level of complexity) [26]. The center closed its doors in April 1990. This privately hired clinic ceased operating on patients covered by social security on 1 January 1986, when the Incompatibilities Law in the public sector was enacted [27]. This decision resulted in a significant reduction in its activity.

### 3.2. Policlínico Médico Quirúrgico Nuestra Señora del Carmen

This hospital was a local facility in Calahorra, was established in 1968, contained approximately fifty beds, and was hired by the Public Health Service of La Rioja. It was founded by Dr. Victorino Imaz Jiménez, Dr. José Piñeiro Esparza, and Dr. Esteban Lana Beaumont. In the first phase (1968–1989), its activity was focused on acute patient care. In the second phase (1990–1997), assistance was reoriented, mainly towards chronic patients. For establishing the agreement, it was classified as a general hospital, with departments for medicine, surgery, obstetrics and gynecology, pediatrics, and emergency services (classified in Group IV, surgical, type IA) [26]. One of the most valued specialties was obstetrics, because for years it was the solution to giving birth in the city for women who lived in this geographical area. There was an upward trend in activity from patients referred by the INSALUD to the polyclinic from 1989 to 1991. Gradually, especially from 1992 onwards, the INSALUD began to limit the referral of chronic patients and long hospitalizations. In April 1997, the existing agreement was denounced with the establishment of the Calahorra Specialties Center, the embryo of the Calahorra Hospital Foundation [28], and this fact partly conditioned the future of the polyclinic, which closed its doors on 30 April 1997.

### 3.3. Policlínico Riojano Nuestra Señora de Valvanera

The hospital was a private center in Logroño belonging to gynecologists Drs. Barquín and Maguregui. It was established in 1969 as a general hospital. From the beginning, it maintained an agreement with the Public Health Service of La Rioja. Initially classified as a non-beneficent and surgical hospital in 1986, it became a general hospital in 2006 (classified in group IV, level IA) [26]. The hospital’s activity gradually focused on convalescence (medium stay), and then it evolved into a social and healthcare center. The polyclinic had a singular agreement. The complementary provision modalities corresponded to the medical and surgical areas [17]. It was acquired by the company Viamed Salud, one of the leading hospital groups in the Spanish market created in 2001, which is owned by MEIF Amistad Holding Sociedad Limitada and belongs to Macquarie Assets Management. In 2013, it moved to a location near Hospital San Pedro. In 2007, the opening of the San Pedro Hospital was completed, with the start-up of the High-Resolution Center for Healthcare Processes, which led to an adjustment in the activity carried out in the polyclinic due to changing healthcare needs. Like the case of the Sanatorio Velázquez, the polyclinic ceased to operate on patients covered by social security on 1 January 1986, when the Incompatibilities Law in the public sector came into effect. This decision resulted in a significant reduction in its activity.

### 3.4. Clínica Los Manzanos

This hospital was built in 2004 in Lardero, a municipality located 5 km from Logroño, by the Viamed Salud Group [20]. It is a general hospital with medical–surgical specialties. It was classified as a non-beneficent private institution with a hospitalization capacity of approximately 50 beds [29]. From its inception, it had an agreement with the Public Health Service of La Rioja. The Public Administration and Finance Department awarded the Management of Public Service agreement to Los Manzanos Clinic for providing healthcare services for surgical and interventional diagnostic and therapeutic procedures [30]. The hospital’s billing to the Health Department showed an upward trend from 2016 to 2018. However, it decreased thereafter until the Government of La Rioja suspended the agreement in 2021.

### 3.5. Centro Sociosanitario de Convalecencia Los Jazmines

The hospital was located in Haro and was established in 2005. The project was carried out by order of a Temporary Union of Companies. The managing entity was FIRSA II-Inversiones Riojanas Sociedad Anónima, which brought together several companies, including Viamed Salud Sociedad Limitada [21]. It was classified as a non-beneficent private hospital specializing in geriatrics and/or long-term care. Healthcare was provided to patients from La Rioja Alta who simultaneously required healthcare and social attention. The center had 140 places, of which 115 were for assisted residents and 25 for Alzheimer’s and other dementia patients. It was built through a public works concession agreement. A social and healthcare agreement was established between the Government of La Rioja and this center for beneficiaries of the National Health Service, the activity of which increased over time. This hospital, owned by the Government of La Rioja, had a dual function: first, it played a role as a convalescence and medium-stay center, attending to patients who suffered cardiovascular accidents, fractures, and chronic illnesses; second, it acted as a nursing home with 109 publicly owned places, 25 dedicated to high-dependency residents and the rest to severe-dependency residents [31].

### 3.6. Centro Sociosanitario de Convalecencia Nuestra Señora Virgen del Carmen

This hospital began its journey in Calahorra in 2013 with 75 beds. The Government of La Rioja awarded Policlínico de Valvanera Society, belonging to Viamed, the construction and management company for this center, through a public works concession agreement lasting for 25 years. The center was classified as a non-beneficent private hospital specializing in geriatrics and/or long-term care. The center’s objective was to provide healthcare to patients from Rioja Baja who simultaneously required healthcare and social attention. The activity and economic amount show an upward trend. The start-up represented a total investment of approximately EUR 7 million and created 70 direct jobs and another 50 indirect jobs. With the creation of this center, the Riojan government completed its network for long-term patient care. Together with the 29 places in the Los Jazmines residence in Haro and the 75 places in the Nuestra Señora de Valvanera, in Logroño, it had 176 places in total [30].

### 3.7. Fundación Hospital Calahorra

Calahorra Hospital Foundation was established by the National Health Institute in 2000. It had a collaboration agreement formalized on 5 April 2001, signed between the Ministry of Health and Social Services of the Government of La Rioja and the hospital. Each year, the Department of Health signed an additional clause. The purpose of the agreement was to provide healthcare to beneficiaries of the Public Health Service of La Rioja [24].

## 4. Discussion

### 4.1. Evolution of Private Hospitals in La Rioja

Throughout the years, there was a promotion of the development of public hospitals and the establishment of a greater number of medical and surgical specialties, aiming to reduce the dependence of specialized hospitals on neighboring Autonomous Communities, which resulted in a decrease in the referral of patients to Navarra, Cantabria, Aragón, and Basque Country, compared with that in the previous period [32]. Additionally, agreements were developed with private hospitals. In general, these agreements were hired between a healthcare facility (private or public) and the organization responsible for managing the social security healthcare services, by which the characteristics and rates of each healthcare service agreement are determined under certain conditions. In some cases, the relationship between the hospital and the managing entity is governed by a linking agreement or a singular agreement [33]. The public and private hospitals in La Rioja adapted in accordance with political decisions and the evolution of the demographic and epidemiological patterns of disease and disability processes, as well as user groups’ expectations and the system’s resources. Here, we focused on privately owned dependency hospitals. Throughout this study, two privately owned general hospitals closed: Sanatorio Velázquez (1965–1990) and Policlínico Médico Quirúrgico Nuestra Señora del Carmen (1968–1997). Some of the factors involved were the changes associated with transformations in disease diagnosis and treatment, surgical methods, and the modernization of medicine in general, particularly hospital care, as they proved demanding in financial, logistical, and human capital terms. The effects of the economic crisis led to a reduction in private demand, the absence of generational family succession, and a reduction in agreements with social security, which undoubtedly contributed to the survival of these hospitals. An example of a change in the mission and adaptation of the service portfolio was Policlínico Riojano Nuestra Señora de Valvanera, which, although born as a general hospital, transformed into a convalescence hospital due to the reduction in agreements and social changes. We observed a process of specialization of private hospitals in the sociosanitary space. As already mentioned, the purpose of sociosanitary agreements was to provide healthcare to patients who, having passed the acute phase, required simultaneous and synergistic attention from health and social services at the Viamed-Haro, Viamed-Valvanera in Logroño, and Viamed-El Carmen de Calahorra centers (Table 2).

### 4.2. Public–Private Collaboration in La Rioja

There is no doubt about the diversity of opinions on the PPP model’s theoretical discourse. On the one hand, the theoretical benefits of the model and the probability of favorable results are presented. In this theoretical framework, PPPs are proposed as win-win arrangements and are distinguished by their long useful life and significant private sector financing [34,35]. There are health sector policymakers that are increasingly proposing PPP projects, in which private companies are contracted by governments to finance and deliver new hospitals and related services [36]. Other studies available argue that PPPs combine the strengths of private partners (e.g., innovation, technical knowledge and skills, management efficiency, and entrepreneurship) with the strengths of public partners (e.g., social responsibility, social justice, public accountability, and local knowledge), thus creating an enabling environment for the provision of high-quality health infrastructure and services [37,38]. Others also point out that competition and collaboration between the public and private sectors is profitable both economically and politically and has led to an increase in the number of private hospitals and clinics. This was the case that occurred in Spain prior to the General Health Law [39]. 

However, there are conflicting results [37], and we observed the dependence on political and economic changes that lead to both successes and failures in this field [37,40]. Another concern identified in the proposed model is the public interest in healthcare provision. The guiding principles of PPP initiatives should be based on the potential benefits to society rather than the mutual benefit to partners and should be centered on the concept of health equity to preserve public interest. Nevertheless, the advantages of the PPP model over the public contracting paradigm are still hotly debated. Cost, quality, flexibility, and complexity are just a some of the pressing concerns that persist [37,41].

Furthermore, it is argued that the main factors and characteristics that lead to the success of the PPP model are still not understood, with an increasing number of problems related to the implementation of this type of model being identified. It can also be considered that PPPs in the health sector should not only fill the gaps in the provision of services to the population and dominate the development of infrastructure allocated to healthcare, but should also offer several other potential benefits, such as by enabling health programs, run by governments, to be accelerated or to promote the motivation of both health professionals and patients [41].

A characteristic element of La Rioja has been the promotion of public–private collaboration. Private initiatives have increased in the Autonomous Community of La Rioja and have been focused on service agreements and concession contracts. From the Government of La Rioja, the arguments put forward to justify public–private collaboration are threefold: financial, efficiency, and risk transfer. In practice, the model of the hospital system in La Rioja is characterized by a predominance of public hospitals and beds over private ones, which increased from 2000 with the opening of Calahorra Hospital Foundation. There was an upward trend in the proportion of private beds during the study period due to the incorporation of two convalescent sociosanitary hospitals. This is reflected in the evolution of the budget allocated to agreements in the Ministry of Health’s budgets. This evolution is presented in La Rioja and at the national level. In 2002, the percentage of healthcare spending dedicated to agreements in Spain quadrupled relative to that of La Rioja (Spain: 10.27; La Rioja: 2.64). The difference decreased throughout the study (2002–2019), although it remained higher in Spain. In 2019, the proportion in Spain was 8.85, and in La Rioja, it was 7.45 (Figure 3) [42].

The chosen formulas for sociosanitary agreements in La Rioja were the Management Services Agreement and the Work Concession. Regarding the Management Services Agreement, on the one hand, in the sociosanitary field, the case of Nuestra Señora de Valvanera was exemplary, with a term of 10 years, which ended in 2024. As for the sociosanitary centers Los Jazmines (Residence with Convalescence Unit) and Nuestra Señora Virgen del Carmen (Convalescence Unit), since they were newly constructed, the Government of La Rioja opted for the Work Concession, in which work and service were implemented. The duration was up to 40 years. When the concession ends, the centers will become the property of the Department of Health (in the case of Nuestra Señora Virgen del Carmen, it expires on 31 December 2037). In La Rioja, until 2004, patients requiring cardiac surgery had to be referred to other Autonomous Communities such as Cantabria. Therefore, once resources were maximized, the competent Health Department of the Government of La Rioja established agreements called “Concerts or Singular Linking Agreements”. On the other hand, a Management Services Contract was established with the Clínica Los Manzanos to provide healthcare for surgical procedures and diagnostic and therapeutic interventional procedures to beneficiaries of the Public Health Service of La Rioja.

This study also has some limitations. Although there was continuity in data collection between 1986 and 2019, obtaining information regarding private hospitals was challenging. Research could be expanded in the future to understand the impact of the COVID-19 pandemic on La Rioja’s healthcare system. Additionally, comparative studies on different hospital management models in Spain and in other European countries could be conducted to analyze their impact on healthcare activity, patient and worker satisfaction, and the region’s economy.

## 5. Conclusions

The hospital system of La Rioja is characterized by a predominance of public beds compared with private ones, although an increasing trend has been observed in the number of private beds since 2013. We observed that the development of the hospital system in La Rioja during the period between 1986 and 2019 was determined by the presence of a mixed public–private hospital system. Having private hospitals facilitated less of a dependence on hospitals of other Autonomous Communities. From 2013 onwards, an increasing trend was observed in the proportion of private beds of 22%, while the proportion of public beds decreased to 78% (a more pronounced pattern than that at the national level) due to the addition of two hospitals dedicated to long-term care stays of a sociosanitary nature.

Private general hospitals were seriously affected by the interruption of the agreements with the Public Health Service. Two of them closed: Sanatorio Velázquez (1965–1990) and Policlínico Médico Quirúrgico Nuestra Señora del Carmen (1968–1997). Other changes came from the transformations in the diagnosis and treatment of diseases, new surgical methods and the modernization of medicine in general, and the need to have more specialized professionals, all of which had a high financial and logistical impact. In addition to this, they suffered the effects of the economic crisis of 2010, which caused a reduction in private demand. Over time, there was also a generational change in the owners and managers of these hospitals.

Some private hospitals, such as Nuestra Señora de Valvanera, reacted by adapting to the needs of the public system’s service portfolio. Thus, this hospital moved from being a general hospital to a convalescent hospital. This institution joined with other private hospitals that occupied the socio-health space. We observed that long-term hospitals are an area in which public–private collaboration has been promoted, which is understood as a strategic alliance.

A characteristic element of the La Rioja health system has been the promotion of public–private collaboration. The formulas chosen for the social and health agreements in La Rioja have been the Service Management Contract and the Work Concession Contract.

This study reveals that the changes that occur in the relationship between public health services and private health institutions are continuous and that it will be the evolution of clinical, social, economic or political issues that will lead to the evolution of these relationships, which, in any case, should preserve equity and access to health care.

## Figures and Tables

**Figure 1 healthcare-12-00990-f001:**
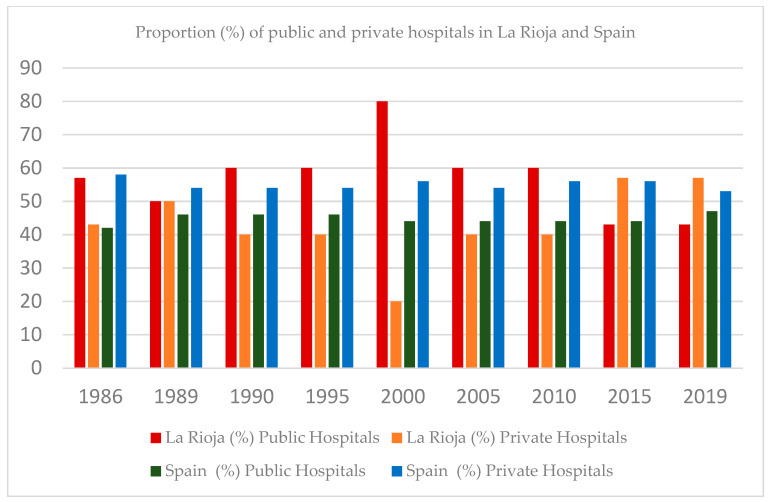
Proportion (%) of public and private hospitals in La Rioja and Spain. Source: national hospital catalogs (1986–2019).

**Figure 2 healthcare-12-00990-f002:**
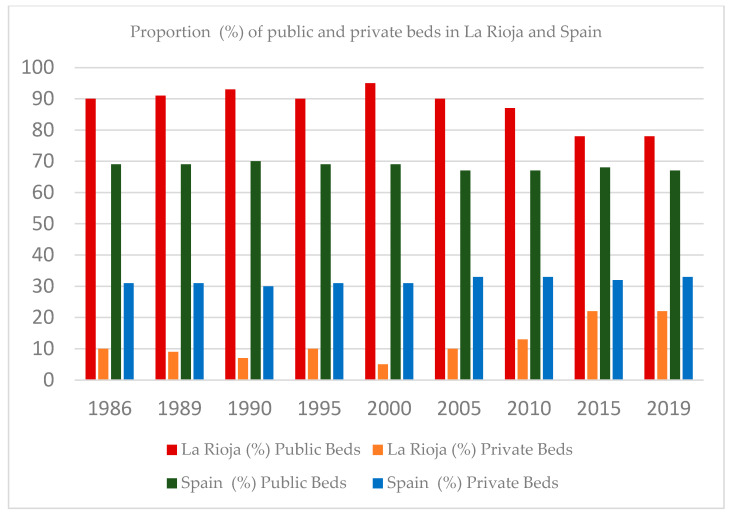
Proportion (%) of public and private beds in La Rioja and Spain. Source: national hospital catalogs (1986–2019).

**Figure 3 healthcare-12-00990-f003:**
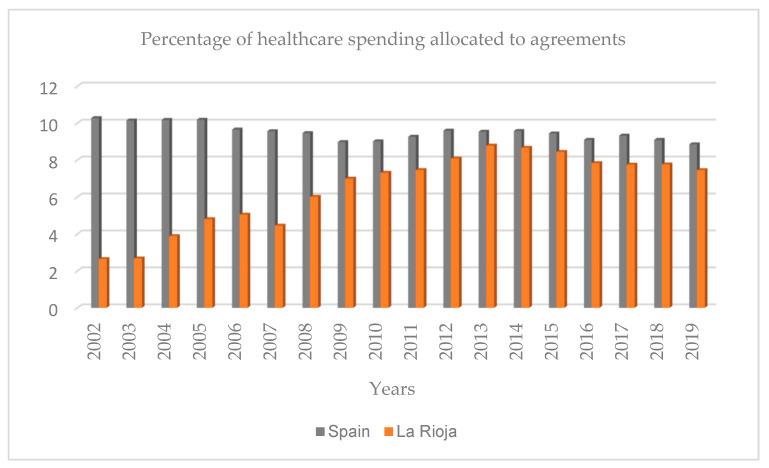
Percentage of healthcare spending allocated to agreements (Spain and La Rioja). Source: public health expenditure statistics; Ministry of Health, 2024.

**Table 1 healthcare-12-00990-t001:** Hospitals in La Rioja (1986–2019).

	General Hospitals	Property Dependency	Temporal Period	Number of Beds
1.	Hospital de La Rioja	Logroño Council Autonomous Community	1986–2004	185
2.	Hospital San Millán	Social Security	1986–1988 1988–2009	540
3.	Hospital San Pedro	Institutional Administration of National Health Social Security	1986–1988 1988–2004 2004–2009 2009–2019	215
4.	Complejo Hospitalario San Millán-San Pedro	Social Security	1988–2004	755
5.	Complejo Hospitalario San Millán-San Pedro de La Rioja	Social Security	2004–2009	940
6.	Fundación Hospital Calahorra	INSALUD Social Security	2000–2019	83
7.	Sanatorio Velázquez	Private	1986–1990	56
8.	Policlínico médico quirúrgico Nuestra Señora del Carmen	Private	1986–1997	47
9.	Policlínico Riojano Nuestra Señora de Valvanera	Private	1986–2004	55
10.	Clínica Los Manzanos	Private	2004–2019	48
	Psychiatric Hospitals	Property Dependency	Temporal Period	Number of Beds
1.	Centro Asistencial Reina Sofía	Autonomous Community	1986–2009	180
2.	Centro Asistencial Albelda de Iregua	Autonomous Community	2009–2019	140
	Convalescence Hospitals	Property Dependency	Temporal Period	Number of Beds
1.	Los Jazmines	Private	2005–2019	26
2.	Nuestra Señora de Valvanera	Private	2004–2019	87
3.	Nuestra Señora Virgen del Carmen	Private	2013–2019	75

Source: national hospital catalogs (1986–2019).

**Table 2 healthcare-12-00990-t002:** Sociosanitary agreements, object and bed resources, and health and social care convalescence hospitals in La Rioja.

Hospital	Type	Object	Bed Resource
Los Jazmines	Public works concession agreement for a Convalescent Socio-Sanitary Center in Haro (upper Rioja).	Healthcare for patients who, once the acute phase has been overcome, require simultaneous and synergistic attention from health and social services.	A total of 26 beds for convalescence
Nuestra Señora de Valvanera	Service management agreement for a Convalescent Socio-Sanitary Center in Logroño (middle Rioja).		A total of 84 beds for convalescence 3 beds for permanent and persistent vegetative state care Total: 87
Nuestra Señora Virgen del Carmen	Public works concession agreement for a Convalescent Socio-sanitary Center in Calahorra (Lower Rioja).		A total of 75 beds for convalescence

Source: Reports from the Ministry of Health and Social Services of La Rioja (2004, 2011, and 2014). Reports from the Ministry of Health of La Rioja (2017, 2018, and 2019).

## Data Availability

Data are contained within the article.

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
