# Peer review of "The Collaboration of Private Hospitals with the Public Health Service: The Case of La Rioja, Spain (1986–2019)"

_healthcare, 2024, doi:10.3390/healthcare12100990_

Round 1

Reviewer 1 Report

Comments and Suggestions for Authors

The authors highlighted an important issue affecting the delivery of healthcare services to Spaniards. However, major changes are needed to render this paper suitable for publication:

1. The abstract does not describe the methodology used. It is confusing to the reader when trying to understand how the study was carried out.

2. The introduction is concise and straight to the point. However, the purpose of the paper should be further clarified at the end of the introduction. Additionally, the sentence "This study began in 1986 with the approval of the General Health Law and ended in 2019 74 to delimit a period of homogeneous healthcare activity without the particularities derived 75 from hospital care during the exceptional situation experienced during the coronavirus 76 SARS-CoV-2 (COVID-19) pandemic" should be moved to the methodology.

3. The main issue with this manuscript is the methodology. What do you mean by documentary study? It is also apparent that data analysis was carried out. Which software was used to analyze such data? Where was the data collected from? What type of analysis was carried out? When looking at the primary and secondary documentary sources, which type of qualitative analysis did you carry out? It seems to be content qualitative analysis carried out over a specific time period. 

4. The methods section needs major re-structuring. It needs to be clear that this is a mixed-methods study using both qualitative and quantitative data, with details on how the analysis was carried out and how the data was collected. The results and discussion will then be adjusted accordingly.

5. In the conclusion, remove and replace the word "thanks". 

6. For the discussion, how will your study inform proposed changes in the healthcare system based on your findings? Can you compare Spain to another European country with a similar healthcare infrastructure?

Author Response

Here the response to your comments.

Thank you.

Reviewer 2 Report

Comments and Suggestions for Authors

Jimenez-Bunuales et al. describe the public and private hospitals systems in La Rioja Spain (1986-2019).

This is a well put together paper with no detectable mistakes, and limitations are discussed well. However, I have two concerns that I think should be addressed.

1. Hospital allocations to public and private beds is dictated by funding - although you do mention this it would improve clarity for the reader and your story, if there was a brief discussion on how your healthcare system functions - is it user pays? do you have Medicare ect? this would help to establish the rationale for your publication. 

2. References - overall you have a limited number ie 23 in total - they are all local, does Spain have a similar healthcare system to anyone else? have they investigated your research question? Where did the idea come from to publish this work? This would help to establish the overall outcome - which is lacking in the conclusion. Is this a scientific research study or a government report? 

Author Response

Following your instructions we have answered your comments.

Thank you.

Reviewer 3 Report

Comments and Suggestions for Authors

Dear authors,

Thank you for your paper. It was interesting to read, however, I think you can improve the paper with minor changes.

Chapter 1

The introduction makes a good historical resume but fails to provide a correct theoretical framework regarding the main topic under investigation: The "public-private collaboration among hospitals". Please include subtopic 1.1 where the authors will explain the PPP theoretical framework and main references. The Authors can find several recent papers about this topic, especially in Portugal where the theme was deeply investigated in the last 2 years.

The GAP is not presented and the aim of the study is not sufficiently explained. The simple "We have reviewed the evolution of hospitals: how and why some were closed and others were opened." is insufficient and does not provide a comprehensive scope to the paper.

Chapter 2

The methodology lacks further explanation and needs references to previous applications. When the Authors state "Additionally, we analyzed data" I suppose it means that they resource to archival research method. Please explain the methods clearly and provide the relevant references. 

Regarding Chapters 3,4 and 5

The paper looks much more like a narrative interpretation of documents and not a research paper. In my opinion, at best this paper could be classified as a Critical review even though it would require extensive corrections.

Comments on the Quality of English Language

I suggest proofreading the document. Some major corrections are necessary. 

Author Response

(The authors gave the same response as above.)

Reviewer 4 Report

Comments and Suggestions for Authors

The study focuses on a global issue that has profoundly impacted governments and populations worldwide. With the escalating pressures of rising hospitalizations and population growth on healthcare systems, there is an urgent need to optimize the utilization of resources, such as limited beds and budgets. Consequently, exploring the potential for collaborative resource utilization between private and public hospitals becomes a significant issue warranting examination. The paper must address the following points:

1. The study's methodology, located in the final paragraph of the Introduction, requires expansion to provide more comprehensive details.

2. An organizational paragraph should be appended to the Introduction to enhance its structure.

3. The absence of a Literature section, a crucial aspect of scientific research, needs rectification. This section should incorporate references focusing on bed and resource management in both public and private healthcare settings. These references should address the challenges of resource reallocation and budget limitations experienced globally, particularly in developed nations. By doing so, the study can serve as a model for countries facing similar issues. The following reference exemplify resource reallocation in healthcare systems: - A Hybrid Analytical Model for an Entire Hospital Resource Optimisation. https://doi.org/10.1007/s00500-021-06072-x

4. The Conclusion section is currently brief and lacks depth. It should offer a comprehensive summary of the study's findings and outline the benefits for various stakeholders, including patients, healthcare providers, hospital administrators, and government health ministries.

5. Additionally, the Conclusion should discuss the potential implications of the study for future research endeavors.

Comments on the Quality of English Language

Minor editing of English language required.

Author Response

(The authors gave the same response as above.)

Round 2

Reviewer 1 Report

Comments and Suggestions for Authors

We thank the authors for addressing the major concerns in the initial review. No additional changes are needed.

Author Response

Thank you!

Reviewer 3 Report

Comments and Suggestions for Authors

Many thanks for the efforts to improve the paper. 

Chapter 1

Despite the improvements presented, and since the Authors are trying to conclude about the benefits of the PPP model, I strongly suggest that the Authors review some recent additional works and include them in the Theoretical Framework. It is not enough to define the “PPP” term, for this kind of work it is mandatory to present the state of the art regarding how the recent literature views the empirical advantages of the PPP model. I strongly suggest these papers:

 https://doi.org/10.1590/0103-6351/2935

 https://doi.org/10.3390/healthcare11121723

 https://doi.org/10.1016/j.heliyon.2023.e19122

 https://doi.org/10.1016/j.socscimed.2014.03.037

Chapter 2

The methodology applied still lacks explanation. The Case Study Methodology is presented but then no specific method is explained or applied. I feel this paper is much more a narrative interpretation of a set of archival documents rather than a research paper. I would like to see the document database listing or at least a summary of the data.

 Chapter 4

The Authors pointed out a disagreement regarding the PPP model application results. However, no recent studies are cited, and no empirical evidence is presented. Also, a certain skepticism is mentioned but no references are provided. Several recent researchers have empirically shown how political and economic changes lead to both successes and failures. I strongly recommend reviewing recent literature.

Author Response

Following your instructions, we have answered your comments.

Thank you!

Reviewer 4 Report

Comments and Suggestions for Authors

The required changes have been made.

Author Response

Thank you!
